# FBANet: Transfer Learning for Depression Recognition Using a Feature-Enhanced Bi-Level Attention Network

**DOI:** 10.3390/e25091350

**Published:** 2023-09-17

**Authors:** Huayi Wang, Jie Zhang, Yaocheng Huang, Bo Cai

**Affiliations:** Key Laboratory of Aerospace Information Security and Trusted Computing, Ministry of Education, School of Cyber Science and Engineering, Wuhan University, Wuhan 430072, China; why_kevin@163.com (H.W.); qjhkzj@126.com (J.Z.);

**Keywords:** depression recognition, feature enhancement, bi-level attention, transfer learning, cross validation

## Abstract

The House-Tree-Person (HTP) sketch test is a psychological analysis technique designed to assess the mental health status of test subjects. Nowadays, there are mature methods for the recognition of depression using the HTP sketch test. However, existing works primarily rely on manual analysis of drawing features, which has the drawbacks of strong subjectivity and low automation. Only a small number of works automatically recognize depression using machine learning and deep learning methods, but their complex data preprocessing pipelines and multi-stage computational processes indicate a relatively low level of automation. To overcome the above issues, we present a novel deep learning-based one-stage approach for depression recognition in HTP sketches, which has a simple data preprocessing pipeline and calculation process with a high accuracy rate. In terms of data, we use a hand-drawn HTP sketch dataset, which contains drawings of normal people and patients with depression. In the model aspect, we design a novel network called Feature-Enhanced Bi-Level Attention Network (FBANet), which contains feature enhancement and bi-level attention modules. Due to the limited size of the collected data, transfer learning is employed, where the model is pre-trained on a large-scale sketch dataset and fine-tuned on the HTP sketch dataset. On the HTP sketch dataset, utilizing cross-validation, FBANet achieves a maximum accuracy of 99.07% on the validation dataset, with an average accuracy of 97.71%, outperforming traditional classification models and previous works. In summary, the proposed FBANet, after pre-training, demonstrates superior performance on the HTP sketch dataset and is expected to be a method for the auxiliary diagnosis of depression.

## 1. Introduction

Major Depressive Disorder (MDD) or depression is a common mental illness characterized by symptoms such as low mood, decreased interest, pessimism, slowed thinking, lack of initiative, poor appetite, and sleep disturbances [1]. Severe cases may even involve suicidal ideation or behavior [2]. According to the World Health Organization (WHO), as of 31 March 2023, about 5% of adults worldwide are afflicted with depression [3], and depression is expected to surpass cardiovascular disease and become the leading cause of disability by 2030 [4]. Therefore, efficient and accurate diagnosis of depression is crucial.

Traditional depression diagnosis methods include symptom questionnaires and psychological tests. Commonly used questionnaires include the Hamilton Depression Rating Scale (HAMD) [5] and the Self-Rating Depression Scale (SDS) [6]. These questionnaires contain multiple items covering various aspects of depressive symptoms, such as low mood, decreased interest, and changes in sleep and appetite. Each item has a different score, with higher total scores indicating more severe depressive symptoms. However, questionnaires may not always measure accurately. For example, the test subjects may provide inaccurate answers or conceal their symptoms deliberately, or may have a different understanding of the questions or provide uncertain answers. The House-Tree-Person (HTP) sketch test [7] is a commonly used psychological test that requires the test subject to sketch a house, a tree, and a person in pencil on white paper. Psychologists analyze the features of the drawings to understand the individual’s mental state and personality traits, allowing them to identify the presence of depressive symptoms. This test can capture personality traits that are difficult to express in words and avoid distortion of the response content in the process of verbalization [8,9], making it a more objective method when compared to questionnaire diagnosis methods.

In recent years, with the development of computer technology and artificial intelligence, more and more studies have begun to explore the use of computer-aided diagnosis methods to recognize depression, such as using computers to recognize facial expressions [10,11], Electroencephalogram (EEG) [12], Electrocardiograms (ECG) [13], and speech [14] to analyze whether the test subject is depressed. The above methods can be roughly divided into three steps: (1) data collection, which uses sensors, cameras, microphones, and other devices to collect physiological data such as facial expressions, EEG, ECG, and speech from test subjects; (2) data processing, which preprocesses and cleans the collected data and performs data transformation and normalization; (3) feature extraction and recognition, which uses machine learning and deep learning algorithms to extract features related to depression from the processed data. Then, the extracted features are fed into feature classifiers such as a Support Vector Machine (SVM) [15] and Fully Connected Networks [16] to obtain classification results. However, the above-mentioned methods require special equipment and testing environments, which results in high data collection costs. Additionally, the interpretation of facial expressions, speech, and other data is subjective due to factors such as individual differences and cultural backgrounds.

This paper uses the HTP sketch for the recognition of depression. Notably, using drawing as a method for screening and analyzing depression has the advantage of being more cost-effective compared to detection technologies such as EEG and ECG. This makes it feasible to implement on a larger scale for depression recognition in institutions such as universities and corporations. In the HTP test, the house reflects the test subject’s associations with family and loved ones, the tree reflects their vitality and perception of the environment, and the person often reflects their self-awareness and relationships with others [8]. For example, an entire tree being drawn in dark black or incomplete people, or roofs and walls that are separated, may indicate that the test subject has a psychological disorder [17]. Existing works [18,19,20] on manual recognition of depression are based on the above methods, and there are also a number of works [17,21] based on machine learning and deep learning methods. However, the method of manually analyzing drawing features requires extensive training by doctors, resulting in high time costs. On the other hand, the methods proposed by [17,21] are characterized by laborious processing steps. Therefore, we propose a one-stage depression recognition method for HTP sketches, which has few process steps and a higher degree of automation. The steps are as follows: (1) the HTP sketch is divided into several patches with overlapping edges; (2) the features of patches and the whole sketch are extracted and fused. (3) Self-Attention and Triplet Attention are used to focus on important features and perform attention fusion; (4) the hybrid attention features are fused with the features of the whole sketch again for feature compensation; (5) the classification head is used to process the feature vector to obtain the classification result. In addition, this paper also uses the traditional CNN, Vision Transformer, and existing works [17,21] for experimental comparison. Our contributions are as follows:A deep learning-based, one-stage depression recognition method (FBANet) for HTP sketches is proposed for the first time. The FBANet comprises three key modules: the Feature Enhancement module, which enhances the network’s feature capture ability; the Bi-Level Attention module, which captures both contextual and spatial information; and the Classification Head module, which obtains the classification results. After simple preprocessing, high-accuracy recognition results can be obtained by feeding the images into FBANet, making it an expected auxiliary diagnostic method for depression.Given the small size of the HTP sketch dataset, transfer learning is employed to improve the model’s accuracy and reduce the risk of overfitting. Specifically, the model is pre-trained on a large-scale sketch dataset and fine-tuned on the HTP sketch dataset. Experimental results demonstrate the superior performance of the proposed model.

## 2. Related Work

### 2.1. Traditional Depression Diagnosis

Traditional depression diagnosis and assessment commonly used a scale method, by asking the test subject to answer a series of questions or complete some tasks, and finally using the score to evaluate the degree and type of depression. The scale method is mainly divided into self-assessment scales and clinical assessment scales, see Table 1. However, the assessment results of these scales may be influenced by subjective factors such as the subjects’ personal preferences or doctors’ lack of experience, which may lead to measurement errors. Additionally, these scales require a relatively long assessment time, resulting in high time costs.

### 2.2. Computer Diagnosis of Depression Based on Physiological Signal

Currently, computer-aided diagnosis methods are commonly used to recognize depression. In the study of depression recognition based on facial expressions, Kong et al. [10] employed classic classification architectures such as Fully Connected Networks [25], VGG [26], and ResNet [27] to extract facial image features and perform binary classification. Zhou et al. [28] proposed DepressNet for learning visually interpretable representations of depression. This network is adapted from ResNet50 and first divides video frames into three overlapping regions (top, middle, and bottom), which are then fed into DepressNet for feature extraction. Finally, the features are merged to predict depression scores. The authors used visualization of the network’s activation maps to explain its attention regions.

In the study of depression recognition based on EEG, Wang et al. [12] first collected EEG sequence data of the partial head region of subjects using a three-electrode EEG acquisition sensor. Considering the small amount of data and to prevent overfitting, the data scale augmentation strategy was applied to obtain EEG sequence data expanded two, four, and eight times. To take advantage of convolution in image processing, the sequences were fused into 2D images and VGG was used to extract image features and perform classification. Deng et al. [29] collected EEG sequence information of five parts of the head region of subjects using HydroCel Geodesic Sensor Net. They cleaned the data by performing preprocessing operations such as data denoising and feature smoothing to improve the recognition accuracy. Then, they designed the SparNet, which employed five-branch SeNet and convolution modules to process the EEG information of the five parts of the head region. Finally, the features were fused, and the prediction probability was obtained by classification head.

In the study of depression recognition based on ECG, Zang et al. [13] first collected ECG signals of the subjects using RM-6280C and then preprocessed the data by denoising and normalization. The processed data were segmented and input to a module that includes one-dimensional convolution, max pooling, and fully connected layers, and the classification result was obtained in the end. Zhang et al. [30] extracted 39 RR interval features [31] from the ECG signals of the subjects and used machine learning classification methods such as K-nearest neighbor (KNN) [32], Support Vector Machine (SVM) [15], and Decision Tree (DT) [33] to classify the selected features. They also employed the backward selection algorithm to select key features and improve the recognition accuracy of the model.

In the study of depression recognition based on audio, Lu et al. [14] proposed a CBAM-based attention mechanism network. The authors collected speech data from subjects in four scenarios: vocabulary reading, short text read, interview and picture description, and removed noise such as coughing and misreading. The Mel Frequency Cepstrum Coefficient (MFCC) features [34] of the speech were extracted as input to the neural network, which used a ResNet and CBAM [35] combined architecture. The results were obtained through a classification head. Sardari et al. [36] proposed an end-to-end Convolutional Neural Network-based Autoencoder (CNN AE) technique to learn highly relevant and discriminative features from raw sequential audio data. Notably, the CNN AE first allowed the encoder to learn the raw speech representation, and then the decoder restored the speech. After unsupervised learning on the training dataset, the raw speech in the test dataset was input into the encoder to obtain speech features and then classified using classifiers such as SVM and Random Forests (RF) [37].

Despite the aforementioned research, physiological signals such as EEG, ECG, and audio have drawbacks such as high collection costs, tedious data preprocessing (denoising, characterizing, etc.), small data volume (with only a few dozen participants), and weak interpretability. In contrast, HTP sketches have the advantages of low collection costs (only requiring paper and pen), simple data preprocessing (normalization of the sketches), relatively large amount of experimental data (1615 HTP sketches), and strong interpretability (the drawing features that the model focuses on can be used for judgments).

### 2.3. Analysis of Depression Recognition Based on HTP Sketch

As a projection test, the drawing test has a history of nearly 100 years in modern psychological research, and the measurement efficiency has been recognized as both scientific and therapeutic [38]. Among the drawing tests, Buck’s HTP test [7] is the most classic and popular. It was mentioned that the house represents the test subject’s psycho-sexual adjustment, contact with reality, and accessibility. For example, a big door may represent an extroverted personality, while a small door may indicate introversion and a lack of interest in socializing. The tree represents the test subject’s felt impression of themselves in relation to their environment. A tortuous and twisted trunk, or broken branches, indicating the test subject’s experience of painful trauma. The person represents the test subject’s self-portraiture. For example, an active person may indicate an energetic and adaptable personality. Therefore, the analytical diagnosis of depression can be carried out according to the characteristics of the HTP sketch.

In recent years, Li et al. [39] proposed 35 HTP drawing features associated with depression or anxiety disorders and confirmed their effectiveness through the Rasch measurement model. Yu et al. [19] evaluated the level of anxiety in prisoners before and after psychological treatment using HTP sketches. Yang et al. [18] utilized HTP sketches to diagnose depression in cancer patients and assessed the effectiveness and accuracy of the HTP test by comparing it with the SDS. Hu et al. [40] used HTP sketches to detect depression in middle school students after the Lushan earthquake. Yan et al. [41] used HTP sketches to detect symptoms of depression in high school students. However, these tests are still manually conducted by doctors based on drawing features, and the automation level is low. Moreover, the sample size of the above experiments is small, only several hundred. Zhang et al. [21] conducted a study where they computed the mean of effective pixel, entropy of effective pixel, and the number of corners in the HTP sketches as features for depression recognition. They employed classifiers such as Support Vector Machines (SVM) and Decision Trees (DT). However, this method is only based on the information of pixels and fails to extract the semantic and spatial information of the sketch; Pan et al. [17] developed an automated testing method using a two-stage algorithm that first uses a model similar to R-CNN [42] to locate the drawing features, then uses the binarization method to process shadow features, and finally merges the features and inputs them into a traditional SVM classifier to obtain the classification results. However, this localization method may be biased, and some important features may not be perceived. In contrast, the approach proposed in this paper is a one-stage method that takes the entire sketch as input. After enhancing its features, the Bi-Level Attention is utilized to extract both the semantic and spatial information of the sketch. This approach offers the advantages of a convenient processing pipeline and comprehensive attention to all parts of the sketch.

### 2.4. Image Classification Models and Attention Mechanisms

Image classification is an important task in computer vision. At present, Convolutional Neural Networks (CNN) [43] and Vision Transformers (ViT) [44] are mainly used for image classification.

The CNN structure is mainly composed of a convolutional layer, a pooling layer, and a fully connected layer. We compare classical CNN architectures such as ResNet [27], InceptionNet [45], EfficientNet [46], MobileNet [47]. ResNet consists of a shortcut connection that skips one or more convolutional modules in the network. By stacking residual units, ResNet can produce very deep neural networks with improved training and generalization performance. InceptionNet contains multiple Inception modules. Each Inception module consists of multiple parallel convolutional pathways that allow the network to learn features of different scales and resolutions and to capture both local and global information in the input data. MobileNet uses depthwise separable convolution instead of traditional convolution to reduce the number of parameters and computations. EfficientNet incorporates Mobile Inverted Bottleneck Convolution (MBConv) [48] and Squeeze-and-Excitation Network (SENet) [35], effectively balancing the relationship between network width, depth, and image resolution.

Dosovitskiy et al. [44] proposed the Vision Transformer (ViT) for image classification tasks, which is the first work to apply the self-attention mechanism to this field. Specifically, the input image is first divided into a set of non-overlapping patches, which are then transformed into vector representations using a Transformer encoder with position encoding and self-attention. In self-attention, each vector is compared to others to compute their similarities, and the weights are assigned based on these similarities. The network then obtains a more contextualized representation by computing the weighted average of these vectors, followed by a classification head to output the probability of each class. In this paper, we compare ViT, Hybrid ViT [44], and Swin Transformer [49] with our model. Hybrid ViT differs from ViT in that it uses ResNet50 to extract image features as input vectors; Swin Transformer introduces a multi-scale window attention mechanism to balance computation efficiency and receptive field size. Specifically, Swin Transformer first performs patch partition and linear embedding to obtain image vectors and then computes multiple Windowed Multi-head Self-Attention (W-MSA) and Shifted Window Multi-head Self-Attention (SW-MSA) to reduce computation and capture information between adjacent patches. Patch merge is used to reduce the resolution of image vectors for multi-scale information.

In addition, some studies have proposed visual attention mechanisms. For example, Hu et al. [50] proposed SENet, which introduces a Squeeze-and-Excitation (SE) module. The SE module first performs global average pooling to obtain global features, learns the importance weights of each channel using two fully connected layers, normalizes the weights using the softmax function to generate an SE vector, and then recombines the original feature map by multiplying it with the SE vector. Woo et al. [35] proposed the Convolutional Block Attention Module (CBAM), which introduces a channel attention module and a spatial attention module to adaptively adjust the channel and spatial dimensions of the feature map. The channel attention module learns the importance weights of each channel, while the spatial attention module learns the importance weights of each spatial position. The CBAM module cascades the channel attention module and the spatial attention module to obtain a finer representation. Misra et al. [51] invented Triplet Attention, which calculates attention for image vectors by computing cross-dimensional interaction information and is a combination of SENet and CBAM with reduced parameters. Considering that Triplet Attention is almost parameterless and combines the advantages of SENet and CBAM, this paper combines Triplet Attention with Self-Attention to make up for the lack of unilateral attention.

In general, this paper combines CNN, ViT, and Triplet Attention, aiming to comprehensively extract sketch information and improve the accuracy of depression classification.

### 2.5. Transfer Learning

Transfer learning aims at improving the performance of target learners on target domains by transferring the knowledge contained in different but related source domains [52]. In the field of computer vision, this method mainly solves the problem of overfitting caused by the small size of the target dataset [53]. In recent years, transfer learning has been widely applied. For example, Oquab et al. [54] used transfer learning to first train a traditional CNN model on the ImageNet dataset [55] and then performed classification on the Pascal VOC dataset [56]. Shehada et al. [57] used transfer learning for facial expression recognition. They first pre-trained the model on the FER2013 dataset [58] and then fine-tuned the model on the CK+ dataset [59] to improve the model’s performance. Shin et al. [60] used transfer learning with a CNN structure for thoraco-abdominal Lymph Node (LN) detection and Interstitial Lung Disease (ILD) classification. Apostolopoulos et al. [61] used transfer learning with several CNN networks for Covid-19 identification based on lung CT images. In this paper, we employ transfer learning by first pre-training the model in a supervised manner on a large-scale sketch dataset and then fine-tuning it on the HTP sketch dataset.

## 3. Methodology

Figure 1 illustrates the overall structure of the FBANet. It mainly contains three parts: Feature Enhancement, Bi-Level Attention and Classification Head. In the initial stage, both local and global features are extracted. Subsequently, Feature Enhancement is proposed, where the local features are fused and concatenated with the global features. Next, the combined features traverse the Bi-Level Attention Block. Ultimately, they are concatenated with the global features and processed by the Classification Head to derive the classification probability.

### 3.1. Feature Enhancement

Considering the sparse nature of sketch strokes, the features obtained solely from the global sketch are not sufficiently prominent. To address this issue, we enhance the features by combing local sketch patches and the global sketch.

Step 1: We first resize the sketch image $S\in R^{H\times W\times 3}$, where H=W. Then, we divide *S* into *P* patches $\left\{S_{1},S_{2},\ldots ,S_{P}\right\}$, where $P\in \left\{5,9\right\}$, see Figure 1. When P=5, the whole sketch image is divided into top left patch, top right patch, bottom left patch, bottom right patch, and center patch. Each patch is a square, and its size occupies 36% of the whole sketch image. For the upper right corner $Pos\left(X_{1},Y\right)$ of the top left patch $S_{1}$ and the upper left corner $Pos\left(X_{2},Y\right)$ of the top right patch $S_{2}$, there is a relation $X_{2}<X_{1}$ and $X_{2}{-X}_{1}\leq \frac{H}{2}$. Width and height of each patch is:(1)$$W_{m},H_{m}=\left\{W,H\right\}\times \sigma$$
where σ=0.6. The upper left coordinate of the center patch is calculated as follows:(2)$$X_{m},Y_{m}=\left\{W,H\right\}\times \frac{1-\sigma }{2}.$$
When P=9, we set σ=0.4, and every patch occupies 16% of the total image. It is worth noting that the patches with edge overlap can maintain the hidden context relationship between adjacent patches. Each patch is resized to 224 × 224 and input into the feature extraction network Stem (in this paper, Stem uses ResNet50) to obtain $F_{L}=\left\{F_{1},F_{2},\ldots ,F_{P}\right\}$, where $F_{i}\in R^{c\times h\times w}$, and then compute the average of $F_{L}$:(3)$$\hat{F_{L}}=\frac{F_{1}+F_{2}+\ldots +F_{P}}{P}.$$

Step 2: Resize the whole sketch image *S* to 224 × 224 and input it into the Stem to obtain $F_{G}\in R^{c\times h\times w}$.

Step 3: Feature $F_{L+G}=\left\{\hat{F_{L}};F_{G}\right\}\in R^{(2c)\times h\times w}$ is obtained by attaching $F_{G}$ to $\hat{F_{L}}$.

Step 4: 1 × 1 convolution is used to adjust the channel numbers of feature $F_{L+G}$ to obtain $F_{w}\in R^{N\times h\times w}$, which is convenient for calculating attention.

### 3.2. Self-Attention

The architecture mainly includes the multi-head attention mechanism and the fully connected layer, see Figure 2. The multi-head attention mechanism is used to calculate the importance between each position in the input sequence, and the fully connected layer is used to perform nonlinear transformation of the sequence.

Considering the use of two attention fusion strategies, we do not use classtoken because of the dimension requirement. At the same time, related experiments are performed in the original paper of ViT [44], and it is verified that the presence or absence of classtoken has little impact on the performance of the model.

Step 1: Transforming the dimension $F_{w}\in R^{N\times h\times w}$ to $\hat{F_{w}}\in R^{N\times \left(hw\right)}$ and adding learnable positional encoding and LayerNorm to $\hat{F_{w}}$, as shown in the following equation:(4)$$\hat{F_{w}}=LN\left(\hat{F_{w}}+E_{pos}\right).$$
Here, layerNorm is employed to normalize the features of the input sequence, which effectively mitigates the internal covariate shift within the model, thereby enhancing its stability. Furthermore, position encoding is used to infuse positional information into the input sequence, aiding the model in capturing and understanding the sequence position information.

Step 2: Performing multi-layer (layer = 1, 2, …, *L*) self-attention calculation and residual connection on $\hat{F_{w}}$. Mapping $\hat{F_{w}}$ into three learnable embeddings $\left\{Q,K,V\right\}\in R^{N\times \left(hw\right)}$, the attention matrix is calculated as follows:(5)$$Attention\left(Q,K,V\right)=Softmax\left(\frac{QK^{T}}{\sqrt{C}}\right)V.$$
Here, the scaling factor $\frac{1}{\sqrt{C}}$ avoids the dot products becoming too large and mitigates the degree of gradient vanishing. Multi-head means that *Q*, *K*, *V* are first divided into several head blocks along the channel dimension and each block performs self-attention calculation independently, as shown in the following equation:(6)$$\begin{array}{c} MHSA\left(\hat{F_{w}}\right)=Concat\left(head_{0},head_{1},\ldots ,head_{n}\right)W^{O}\\ \;\;head_{i}=Attention\left(\hat{F_{w}}W_{q}^{i},\hat{F_{w}}W_{k}^{i},\hat{F_{w}}W_{v}^{i}\right) \end{array}$$
where $head_{i}\in R^{N\times \frac{\left(hw\right)}{n}}$ is the output of the *i*th attention head and $W_{i}^{q},W_{i}^{k},W_{i}^{v}\in R^{hw\times \frac{hw}{n}}$ correspond to the input mapping weights. $W^{O}\in R^{hw\times hw}$ is used to map all the heads. The general formula is as follows:(7)$$\hat{F_{w}}=\hat{F_{w}}+MHSA\left(\hat{F_{w}}\right).$$

The purpose of using multi-head self-attention is to allow the model to focus on information from different representation subspaces. Taking the HTP sketch as an example, one of the heads may focus on the style of the drawings from a global perspective, and another head may pay attention to drawing details, such as the thickness and trembling of strokes, which are crucial for recognizing depression.

Step 3: Performing LayerNorm $\hat{F_{w}}$ and feed it into the MLP module with residual connection:(8)$$\hat{F_{w}}=\hat{F_{w}}+MLP\left(LN\left(\hat{F_{w}}\right)\right).$$

The MLP module contains two fully connected layers. The residual connection is used to pass the information of the input sequence directly into the next block. This can effectively accelerate the convergence of the model, avoid the vanishing gradient, and improve the generalization performance of the model. In this paper, *n* = 8 and L = 12.

### 3.3. Triplet Attention

Triplet Attention is a three-branch structure that calculates attention weights along the C, H, and W dimensions and averages them. It can capture interdimensional interaction information in images and has the advantage of having a small number of parameters. The structure of the Triplet Attention Block is illustrated in Figure 3.

Consider the input vector $F_{w}\in R^{N\times h\times w}$, Zpool will calculate the global maximum and average along the dimension $D\in \left\{N,h,w\right\}$ and then concatenate them along the dimension D to obtain a spatial attention tensor of 2×h×w, as shown in the following equation:(9)$$Zpool=\left\{MaxPool_{d}\left(F_{w}\right);AvgPool_{d}\left(F_{w}\right)\right\}.$$
Here, the Zpool operation is able to retain rich feature information while reducing the channel depth to make computation lighter.

In the first branch, the interaction between the *h* and *w* dimensions is established: no dimension transformation is needed, and the calculation is as follows:(10)$$F_{w}^{1}=F_{w}⨀Sigmoid\left(BN\left(Conv\left(Zpool\left(F_{w}\right)\right)\right)\right).$$
Here, Conv represents the convolution operation, which can effectively extract spatial information. The convolution kernel size is 7×7, and padding is used to keep the input and output dimensions the same. Batch Normalization (BN) is applied for normalization purposes. Following this, the attention weights are derived via the Sigmoid function, and the element-wise product operation is performed with $F_{w}$, resulting in the output $F_{w}^{1}\in R^{N\times h\times w}$.

In the second branch, the interaction between *w* and *N* dimensions is established by performing the dimension transformation $F_{w}\to F_{w}^{{}^{\prime }}\in R^{h\times N\times w}$. The calculation process is the same as Equation (Equation 10), and the result is $F_{w}^{2^{{}^{\prime }}}\in R^{h\times N\times w}$. Then, the dimension is restored: $F_{w}^{2^{{}^{\prime }}}\to F_{w}^{2}\in R^{N\times h\times w}$.

In the third branch, the interaction between *h* and *N* dimensions is established by performing the dimension transformation $F_{w}\to F_{w}^{{}^{\prime }}\in R^{w\times h\times N}$. The calculation process is the same as Equation (Equation 10), and the result is $F_{w}^{3^{{}^{\prime }}}\in R^{w\times h\times N}$. Then, the dimension is restored: $F_{w}^{3^{{}^{\prime }}}\to F_{w}^{3}\in R^{N\times h\times w}$. Later, averaging $F_{w}^{1},F_{w}^{2},F_{w}^{3}$:(11)$$\bar{F_{w}}=\frac{F_{w}^{1}+F_{w}^{2}+F_{w}^{3}}{3}.$$

### 3.4. Bi-Level Attention Fusion and Classification Head

The calculated channel attention and self-attention are connected, and finally the global feature map is attached to make up for the lack of details in the attention calculation:(12)$$F=Concat(Concat\left(\hat{F_{w}},\bar{F_{w}}\right),Conv\left(F_{G}\right))$$
where $F\in R^{N\times h\times w}$, note that $\hat{F_{w}}$ requires dimension conversion $R^{N\times (hw)}\to R^{N\times h\times w}$, Conv stands for convolution operation with 1 × 1 kernel size.

The Classification Head consists of three blocks: 1 × 1 Convolution (Conv), Global Average Pooling (GAP), and Fully Connected Layer (Linear). The formula is as follows:(13)Output=Linear(GAP(Conv(F)))
where $Output\in R^{1\times num\_\mathrm{class}}$ and num_class represents the total class numbers of dataset. The 1 × 1 convolution is used to adjust the number of channels and reduce the amount of calculation; global average pooling plays a pivotal role in summarizing spatial information without any trainable parameters. Finally, a fully connected layer is used to output the classification probability.

## 4. Experiments

At present, the number of HTP sketches is small, only about 1600, and the attention mechanism network needs a large number of training samples to better fit the distribution of data. Therefore, we use the transfer learning strategy. Firstly, the model is pre-trained in a supervised form on a large-scale sketch dataset, then it is transferred to the HTP sketch dataset for fine-tuning. In addition, we select several classical CNN and Transformer models for comparison, and the training and testing of all models are carried out under the same hyperparameters and environment. The configurations of FBANet are shown in Table 2.

### 4.1. Datasets and Settings

#### 4.1.1. Datasets

Pre-training experiments are conducted on the QuickDraw-414K dataset [62]. QuickDraw-414K is randomly selected from the QuickDraw dataset [63], which contains about 50 million sketches. Specifically, the dataset consists of 345 classes, each with 1000 training samples, 100 validation samples, and 100 test samples, with a resolution of 224 × 224 pixels. Considering that the sketches in the QuickDraw-414K dataset are black backgrounds with white strokes, contrary to the white background with black strokes in the HTP sketch dataset, color conversion is also necessary. Figure 4 shows several examples of the QuickDraw-414K dataset.

Fine-tuning experiments are conducted on the HTP sketch dataset, which is sourced from the work of Zhang et al. [21] and is continuously updated. Currently, a total of 1615 test subjects participated in the study, including 1296 normal individuals and 319 depressed individuals. Each test subject drew only one sketch. Therefore, the HTP sketch dataset now consists of a total of 1615 sketches, with 1296 drawn by healthy individuals and 319 drawn by depressed individuals. Each sketch has a resolution of 4676 × 3308 pixels. Figure 5 shows several examples of the HTP sketch dataset.

In Figure 5, differences between the drawings of individuals with depression and those of healthy individuals are observed. In Figure 5a,b, the brush strokes exhibit moderate pressure, the lines appear smooth, and the overall style is normal. In Figure 5c, the presence of falling raindrops, withered trees, and single-line trunks symbolically represents a state of low mood and depression. In Figure 5d, heavy brush strokes, dark tree trunks, disorderly branches, a hanging angel, falling tears, cracked walls, and an overall strange style indicate that the test subject is suffering from severe psychological depression.

#### 4.1.2. Implementation Details

In the pre-training experiment, we train the FBANet model and the comparative models for 50 epochs in total, using the SGD optimization algorithm and giving an initial learning rate of 3 × 10^−2^. The learning rate update strategy uses the cosine annealing algorithm with Warmup, where the number of Warmup steps is set to 1 epoch. The input sketches are resized to 224 × 224, and batch size is set to 40.

In the fine-tuning experiment, we use five-fold stratified cross-validation to train and validate the FBANet model and train each fold for 10 epochs. The SGD optimization algorithm is employed with an initial learning rate of 1 × 10^−3^ The learning rate update strategy uses the cosine annealing algorithm with Warmup, where the number of Warmup steps is set to 1 epoch. The input sketches are resized to 224 × 224, the batch size is set to 16, and the parameters of the model are not frozen. To prevent overfitting and improve generalization, we employ the data augmentation toolkit Albumentations [64,65] to perform data augmentation operations. Specifically, for the training segment data in cross-validation, we apply data augmentation operations such as random horizontal and vertical flips, as well as normalization. For the validation segment data, we only perform normalization operation.

We use the cross-entropy loss function to train the model:(14)$$CrossEntropyLoss=-\frac{1}{N}\sum_{n=1}^{N}\sum_{i=1}^{k}{y_{i}}^{t}\log{y_{i}}^{p}$$
where *N* is the total number of samples, *k* is the number of classes, ${y_{i}}^{t}$ is the class label, and ${y_{i}}^{p}$ is the predicted value of the model.

### 4.2. Metrics

We choose Accuracy, F1 score, Precision, and Recall as the metrics for classification, which are calculated by the symbols of the confusion matrix: True Positive (TP), True Negative (TN), False Positive (FP), False Negative (FN).

Accuracy is the proportion of examples that the model predicts correctly. It is one of the most commonly used evaluation metrics, especially when the distribution of positive and negative samples is relatively balanced. It is calculated as follows:(15)$$Accuracy=\frac{TP+TN}{TP+TN+FP+FN}.$$

Precision is a metric that measures the proportion of true positive samples among all the samples predicted as positive by the model. This metric focuses on how accurately the model predicts positive samples, especially if FP is high. The formulation is:(16)$$Precision=\frac{TP}{TP+FP}.$$

Recall is a measure that quantifies the ability of a model to correctly recognize positive samples from the entire set of positive samples in the dataset. This metric focuses on the ability of the model to recognize positive samples, especially when FN is high. The formulation is:(17)$$Recall=\frac{TP}{TP+FN}.$$

F1 score is the harmonic mean of Precision and Recall, and it takes into account the performance of both Precision and Recall to provide a more comprehensive assessment of the overall performance of the model. It is calculated as follows:(18)$$F1\;score=2\times \frac{Precision\times Recall}{Precision+Recall}=\frac{2TP}{2TP+FP+FN}.$$

### 4.3. Pre-Training

Table 3 shows the results of training and testing on the QuickDraw-414k dataset. The comparative models based on CNN are ResNet50 [27], Inceptionv3 [45], MobileNetv3 [47], and EfficientNetb5 [46].

On the test dataset, MobileNetv3 performs better than other CNN models, with an accuracy of 70.64% and an F1 score of 70.58%. The Transformer-based models used are ViT [44], Hybrid ViT [44], and Swin Transformer [49], with Hybrid ViT performing best with an accuracy of 72.04% and an F1 score of 71.95%. Given that our model combines channel attention with self-attention, it is better than traditional CNN or ViT models. Our FBA-Base-9 accuracy reaches 73.83% and F1 score reaches 73.79%, which is 3.19% and 1.79% higher than MobileNetv3 and Hybrid ViT, respectively.

In FBA models of the same scale, it is generally observed that the more the number of patches, the higher the accuracy. For instance, the FBA-Small-9 model achieves a higher accuracy (73.35%) than the FBA-Small-5 model (70.53%), possibly because the strokes in the images of the QuickDraw-414k dataset are uniform, with sparse and uniform feature distributions. As a result, more patches can capture local details; when the number of patches is the same, it is found that the accuracy increases when the number of attention layers is increased from 6 to 12. For example, the FBA-Base-5 model achieves a higher accuracy (73.81%) than the FBA-Small-5 model (70.53%). However, when the number of attention layers is increased from 12 to 16, the accuracy decreases. For example, the FBA-Large-5 model achieves a lower accuracy (73.21%) than the FBA-Base-5 model (73.81%). This phenomenon may be caused by attention redundancy.

### 4.4. Fine-Tuning

Next, we fine-tune different pre-trained models on the HTP sketch dataset, see Table 4. In the CNN models, ResNet50 achieves the best performance, with an average accuracy of 86.56% and a highest accuracy of 92.26%. In the Transformer models, Hybrid ViT achieves the highest accuracy, with an average accuracy of 88.11% and the highest accuracy of 92.26%. In addition, a comparison is made between our FBANet model and the methods proposed by Pan et al. [17] and Zhang et al. [21]. It is found that their approaches exhibit inferior performance to our model, with accuracies of 91.33% and 85.55%, respectively, whereas our model achieved an accuracy of 97.71% (FBA-Large-5).

Furthermore, we investigate the effect of patch numbers and attention layers in FBANet models of the same scale. We observe that using 5 patches for feature enhancement generally outperformed using 9 patches (FBA-Small-5 96.72% vs. FBA-Small-9 94.49%). This phenomenon can be attributed to the uneven distribution of strokes in the HTP sketch dataset. If the patches are too small (9 patches), some of them may not contain stroke features, leading to a decrease in model performance. On the other hand, when the number of patches is kept the same, we found that increasing the number of attention layers generally improves the model performance (FBA-Small-9 94.49% vs. FBA-Base-9 97.09% vs. FBA-Large-9 97.13%). This is because the HTP sketch dataset contains more features and information, and stacking multiple attention layers can better capture various aspects of the sketch information.

In addition, we present the average confusion matrix of the FBANet on the HTP validation dataset in Figure 6. It is generally observed that the model achieves higher recognition accuracy for depression than for non-depression. This finding suggests that the FBANet is more proficient in detecting depression-related features from the HTP sketches. Furthermore, we find that our models are more prone to misclassify sketches that originally belong to the normal category as depression. For instance, in the FBA-Small-5 confusion matrix, the probability in the upper right corner is higher than that in the lower left corner (0.063 > 0.025). This phenomenon can be attributed to the class imbalance in the HTP sketch dataset.

To further investigate the interpretability of the FBANet, we employ the Grad-Cam algorithm [66] to analyze the regions of interest of the FBA-Large-5 model on the HTP sketch dataset, as shown in Figure 7. In Figure 7, we observe that for (a,b), the model pays attention to all three objects (house, tree, and person) relatively evenly, with focus on the branches, the middle of the house, and the upper body or the whole person. For (c), the model concentrates more on the raindrops and the withered tree. For (d), the model mainly focuses on the disorderly branches, cracks in the wall, and the hanging angel. It can be seen that the model can accurately capture key features.

### 4.5. Ablation Study

In this study, we evaluate the impact of different components in our model on the QuickDraw-414k dataset and HTP sketch dataset. Specifically, we investigate the effects of various components in the FBA-Base-5 model, including (1) the Feature Enhancement module, (2) the Triplet Attention module, and (3) the Self-Attention module. The results are presented in Table 5 and Table 6.

On the QuickDraw-414k dataset, the accuracy of the model decreases by 2.27% (72.34%) when the Feature Enhancement component is removed compared to the Baseline model. Similarly, when the model contains the Feature Enhancement and Self-Attention components, the accuracy decreases by 1.47% (71.54%) compared to the Baseline model. Finally, when the model contains the Feature Enhancement and Triplet Attention components, the accuracy decreases by 1.97% (71.84%) compared to the Baseline model.

On the HTP sketch dataset, the accuracy of the model decreases by 7.37% (89.35%) when the Feature Enhancement component is removed compared to the Baseline model. Similarly, when the Triplet Attention component is removed, the accuracy decreases by 2.91% (93.81%) compared to the Baseline model. Finally, when the model contains the Feature Enhance and Triplet Attention components, the accuracy decreases by 9.17% (87.55%) compared to the Baseline model.

The ablation experiments demonstrate that the Feature Enhancement, Triplet Attention, and Self-Attention components are all effective, and each component is valid for classification performance on the HTP sketch dataset.

### 4.6. Limitations

Although the proposed method has achieved promising results on the HTP sketch dataset, there are still the following limitations:The accuracy of the FBANet models in recognizing the category of non-depression is lower compared to that of recognizing depression. As shown in Figure 6, except for Confusion Matrixes (c,f), where the classification accuracy is almost equal, the remaining Confusion Matrixes exhibit noticeably higher accuracy in recognizing depression. Therefore, future research will focus on improving the accuracy of the models in recognizing the category of non-depression.The FBANet models have a high number of parameters and computational complexity, as evident from Table 4: FBA-Small-5 compared to ResNet50, Inceptionv3, EfficientNetb5; FBA-Base-5 compared to ViT, Hybrid ViT, Swin. Therefore, future research will explore the design of lightweight models for depression classification.

## 5. Conclusions

This article proposes a novel one-stage method for recognizing depression in HTP sketches based on deep learning. Specifically, we design a recognition model, FBANet, based on channel attention and self-attention mechanisms to automatically extract and analyze features from HTP sketches and directly output classification results. Given the limited size of the HTP sketch dataset (only 1615 samples), we employ a transfer learning strategy by pre-training the model on the large-scale QuickDraw-414k dataset and fine-tuning it on the HTP sketch dataset. The findings indicate that our proposed model outperforms traditional classification models and previous works, as it achieves higher accuracies. Specifically, FBANet achieves a maximum accuracy of 73.83% on the QuickDraw-414k test dataset and an average accuracy of 97.71% with a maximum accuracy of 99.07% on the HTP validation dataset. Additionally, our ablation experiments confirm the effectiveness of FBANet. These results suggest that our designed method for recognizing depression in HTP sketches has the potential to serve as an auxiliary diagnostic tool for depression.

In the future, our research will focus on improving the recognition accuracy of the models in the non-depression category and exploring the design of lightweight depression classification models. These two points will enable better practical application of the models in real-world scenarios.

## Figures and Tables

**Figure 1 entropy-25-01350-f001:**
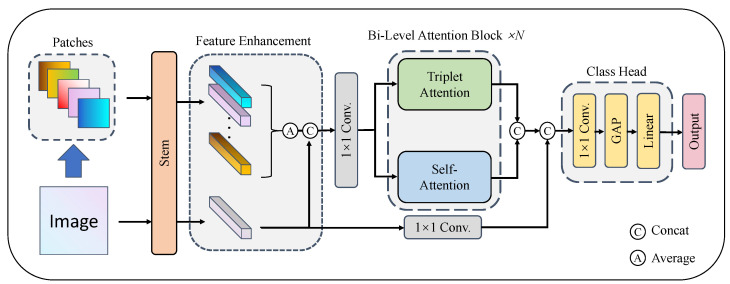
Overall architecture of FBANet. It consists of the Feature Extraction module (Stem), Feature Enhancement module, Bi-Level Attention Block module, and Classification Head module.

**Figure 2 entropy-25-01350-f002:**
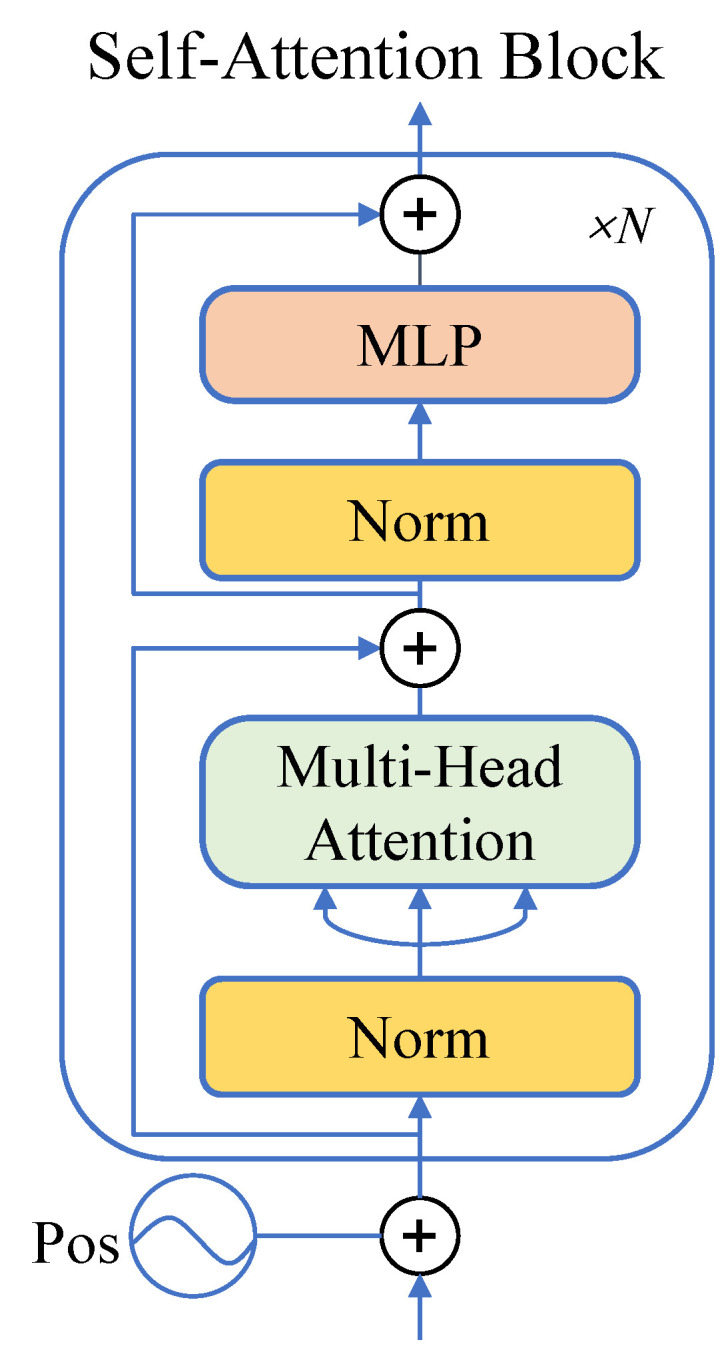
Architecture of Self-Attention Block. It consists of Positional Encoding, Normalization, Multi-Head Attention, Multi-Layer Perceptron and Residual Connection. The whole computation process is repeated *N* times.

**Figure 3 entropy-25-01350-f003:**
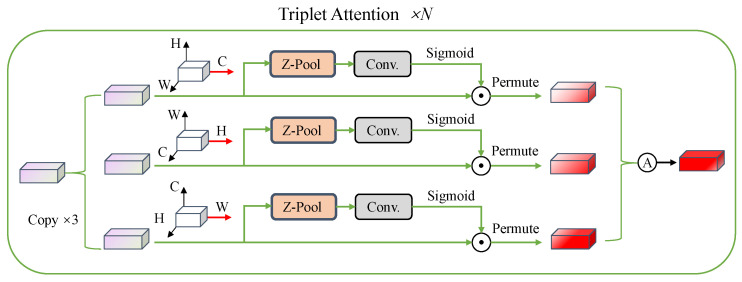
Overall architecture of Triplet Attention. The channel attention is calculated along the C, H, and W dimensions (implemented by Zpool and Convolution modules), so as to capture the interaction information between different dimensions, and finally the three-direction attention fusion is performed. The whole computation process is repeated *N* times.

**Figure 4 entropy-25-01350-f004:**
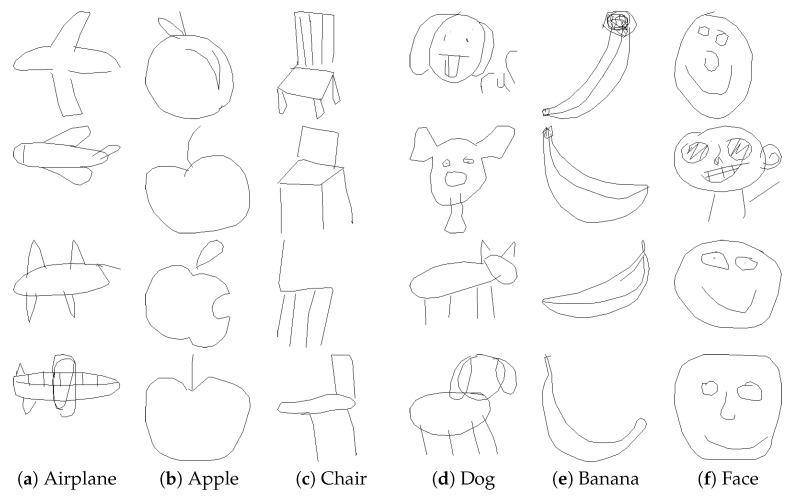
Examples of QuickDraw-414K. We randomly sampled six categories of sketches from the QuickDraw-414K dataset for illustration.

**Figure 5 entropy-25-01350-f005:**
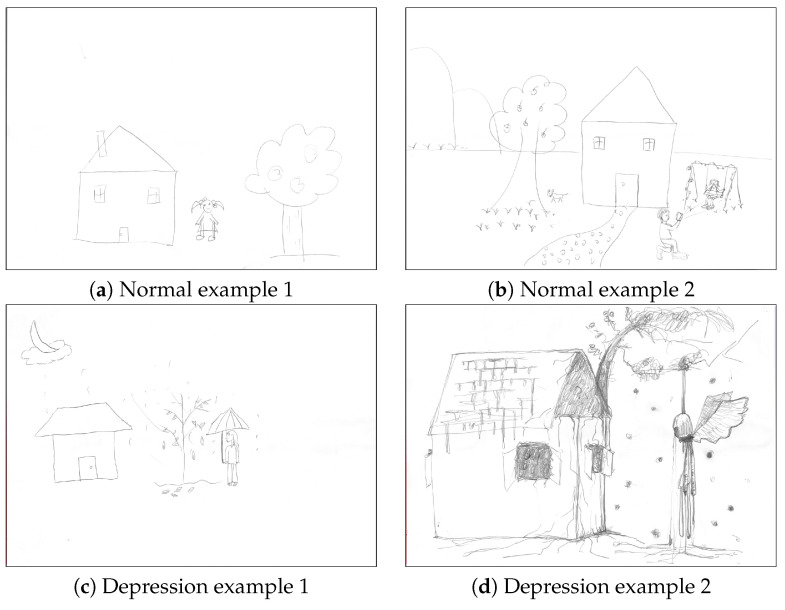
Examples of the House-Tree-Person dataset. We sampled four representative sketches from the HTP dataset to show. Drawing style from sketches (**a**,**b**) is normal. Drawing style of sketches (**c**,**d**) is depressing.

**Figure 6 entropy-25-01350-f006:**
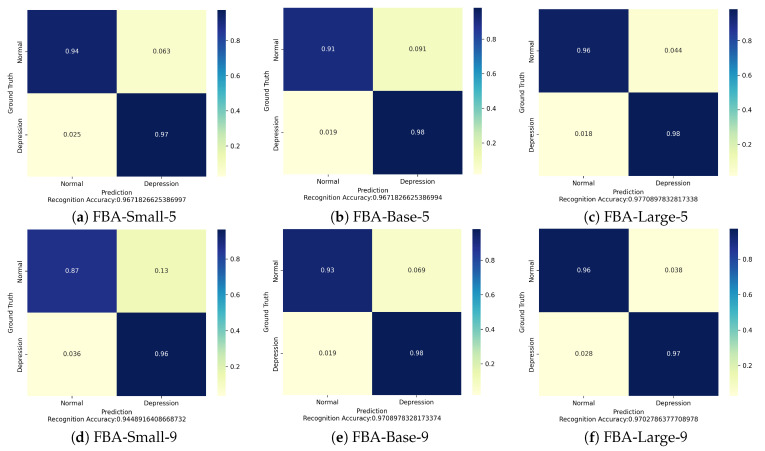
Confusion matrixes of six FBANet models. We show the confusion matrixes of the six FBANet models in the validation dataset of the HTP sketch dataset, comprehensively reflecting the performance of the models in predicting two different categories.

**Figure 7 entropy-25-01350-f007:**
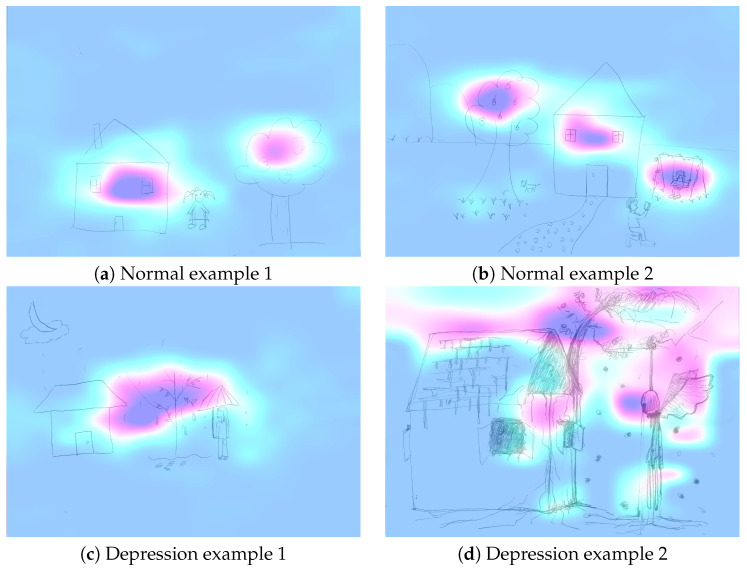
Grad-Cam visualization of FBA-Large-5 model. To illustrate the interpretability of the model, we conduct experiments using the example sketches in Figure 5 to show the important regions that the FBANet-Large-5 model focuses on.

**Table 1 entropy-25-01350-t001:** Details of several self-rating scales and clinical scales.

Kind	Name	Description	Examples	Scale of Scores
Self-Rating Scale	Self-Rating Depression Scale (SDS) [6]	The SDS contains 20 statements, each with 4 different degrees of answer, 0 (rarely), 1 (sometimes), 2 (often), 3 (almost always), corresponding to a score of 1, 2, 3, and 4.	1. I feel sad or depressed. 2. I feel a loss of interest or fun. 3. I feel anxious or scared.	0–52: Normal. 53–62: Mild depression. 63–72: Moderate depression. 73–80: Severe depression.
	Beck Depression Inventory (BDI) [22]	The BDI contains 21 statements, each with 4 different degrees of answer, which are 0 (none or very few), 1 (sometimes), 2 (quite a lot), 3 (extremely severe), corresponding to a score of 0, 1, 2, and 3.	1. Lose interest. 2. Feeling lonely. 3. Feel disappointed.	0–13: Normal. 14–19: Mild depression. 20–28: Moderate depression. 29–63: Severe depression.
	Symptom Checklist-90 (SCL-90) [23]	The SCL-90 contains 90 statements, and there are 13 statements that measure depression. Each statement has 5 different degrees of answer: 1 (never), 2 (very mild), 3 (moderate), 4 (quite a lot), and 5 (severe), corresponding to a score of 1, 2, 3, 4, and 5.	1. Feel your energy levels drop and your activities slow down. 2. Wanting to end your life. 3. You feel lonely.	13–26: Mild depression. 39–65: Severe depression.
Clinical Scale	Hamilton Depression Rating Scale (HAMD) [5]	The original HAMD contains 21 items with 3–5 descriptions for each item, and subjects are required to choose the answer that best fits their situation.	“Depressed mood”: 0. Not present (none); 1. Tell only when asked (lightly). 2. Describe spontaneously in the interview (moderate). 3. The emotion can be expressed without words in the expression, posture, voice, or the desire to cry (severe). 4. The patient’s spontaneous verbal and non-verbal expressions (expressions, movements) almost exclusively reflect this emotion (extremely severe).	0–7: Normal. 8–17: May have depression. 18–24: Depression. >24: Severe depression.
	Hamilton Anxiety Rating Scale (HAMA) [24]	The HAMA contains 14 statements, each with 5 different levels of answers, which are 0 (no symptoms), 1 (mild), 2 (moderate), 3 (severe), 4 (extremely severe), corresponding to scores of 0, 1, 2, 3, and 4.	1. Insomnia. 2. Memory or attention disorders. 3. Depression.	0–7: Normal. 8–14: Mild anxiety symptoms. 15–21: Moderate anxiety symptoms. ≥22: Severe anxiety symptoms.

**Table 2 entropy-25-01350-t002:** Details of FBANet models with different scales.

Model	Layers	Patches	Params (M)	FLOPs (G)
FBA-Small-5 ^1,2^	6	5	58.97	9.16
FBA-Base-5	12	5	101.50	17.52
FBA-Large-5	18	5	144.03	25.87
FBA-Small-9	6	9	58.97	9.16
FBA-Base-9	12	9	101.50	17.52
FBA-Large-9	18	9	144.03	25.87

^1^ ‘Small’, ‘Base’, and ‘Large’ represent the number of attention layers. ^2^ ‘5’ and ‘9’ represent number of patches.

**Table 3 entropy-25-01350-t003:** Performances of FBANet and comparative models on the QuickDraw-414k Dataset.

Model	Accuracy (%)		F1 Score (%)		Precision (%)		Recall (%)		Flops (G)	Params (M)
	**Validation**	**Test**	**Validation**	**Test**	**Validation**	**Test**	**Validation**	**Test**		
ResNet50	69.33	69.53	69.11	69.29	70.12	70.22	69.33	69.53	4.21	24.21
Inceptionv3	69.08	68.92	68.90	68.73	69.27	69.05	69.08	68.92	2.85	25.82
MobileNetv3	70.39	70.64	70.25	70.58	70.48	70.83	70.38	70.64	227.52	4.64
EfficientNetb5	69.93	69.69	69.75	69.53	70.05	69.77	69.93	69.70	2.33	29.05
ViT	67.94	67.90	67.82	67.74	68.21	68.10	67.94	67.90	16.86	86.06
Hybrid ViT	**71.78**	**72.04**	**71.73**	**71.95**	**72.30**	**72.48**	**71.78**	**72.04**	16.91	98.16
Swin	56.75	56.77	56.29	56.28	56.65	56.64	56.75	56.77	15.51	87.1
FBA-Small-5	70.81	70.53	70.68	70.42	71.27	71.03	70.81	70.53	9.16	58.97
FBA-Small-9	**73.43**	**73.35**	**73.34**	**73.24**	**73.73**	**73.56**	**73.44**	**73.35**	9.16	58.97
FBA-Base-5	73.93	73.81	73.91	73.79	74.21	74.09	73.93	73.81	17.52	101.50
FBA-Base-9	**74.01**	**73.83**	**73.98**	**73.79**	**74.27**	**74.11**	**74.01**	**73.83**	17.52	101.50
FBA-Large-5	73.01	73.21	72.96	73.15	73.23	73.42	73.01	73.21	25.87	144.03
FBA-Large-9	**73.79**	**73.75**	**73.76**	**73.75**	**74.01**	**74.10**	**73.79**	**73.75**	25.87	144.03

**Table 4 entropy-25-01350-t004:** Performances of FBANet and comparative models in Horse-Tree-Person (HTP) sketch dataset.

Model	Accuracy (%)		F1 Score (%)		Precision (%)		Recall (%)		Flops (G)	Params (M)
	${\mathrm{Val}}_{\mathrm{Average}}$	${\mathrm{Val}}_{\mathrm{Max}}$	${\mathrm{Val}}_{\mathrm{Average}}$	${\mathrm{Val}}_{\mathrm{Max}}$	${\mathrm{Val}}_{\mathrm{Average}}$	${\mathrm{Val}}_{\mathrm{Max}}$	${\mathrm{Val}}_{\mathrm{Average}}$	${\mathrm{Val}}_{\mathrm{Max}}$		
ResNet50	86.56	92.26	64.97	82.52	72.28	92.86	88.67	100	4.21	24.21
Inceptionv3	82.97	86.69	53.19	65.22	64.18	93.33	61.73	73.44	2.85	25.82
MobileNetv3	85.88	92.26	62.10	80.31	65.50	80.95	65.73	81.25	227.52	4.64
EfficientNetv5	85.26	91.95	62.15	69.92	64.36	78.38	67.92	90.63	2.33	29.05
ViT	82.17	84.21	89.66	90.50	83.33	87.73	**99.92**	**100**	16.86	86.06
Hybrid ViT	**88.11**	**92.26**	**92.75**	**95.06**	**91.50**	**94.66**	98.53	99.61	16.91	98.16
Swin	80.56	81.42	89.19	89.66	80.54	81.25	**99.92**	**100**	15.51	87.1
Pan et al. [17]	85.55	93.33	-	-	-	-	-	-	-	-
Zhang et al. [21]	**91.33**	**95.00**	**91.30**	**95.65**	**95.12**	**97.06**	**87.84**	**94.29**	-	-
FBA-Small-5	**96.72**	97.21	**97.95**	**99.23**	**99.27**	**100**	98.84	**100**	9.16	58.97
FBA-Small-9	94.49	**98.45**	96.54	99.03	97.98	99.61	**99.92**	100	9.16	58.97
FBA-Base-5	96.72	98.45	97.97	99.04	**99.31**	100	**99.23**	100	17.52	101.50
FBA-Base-9	**97.09**	**99.07**	**98.19**	**99.42**	99.11	**100**	98.77	**100**	17.52	101.50
FBA-Large-5	**97.71**	**99.07**	**98.56**	**99.42**	**99.30**	**100**	**99.54**	**100**	25.87	144.03
FBA-Large-9	97.13	98.76	98.12	99.22	99.08	100	98.85	100	25.87	144.03

**Table 5 entropy-25-01350-t005:** Ablation study on the QuickDraw-414k dataset using the FBA-Base-5 model.

Model	Feature Enhance	Triplet Attention	Self-Attention	Accuracy (%)	FLOPs (G)	Params (M)
FBANet		✓	✓	72.34	17.36	100.72
FBANet	✓	✓		71.84	0.72	15.71
FBANet	✓		✓	71.54	17.37	100.91
FBANet	✓	✓	✓	73.81	17.52	101.50

**Table 6 entropy-25-01350-t006:** Ablation study on the HTP sketch dataset using the FBA-Base-5 model.

Model	Feature Enhance	Triplet Attention	Self-Attention	Average Accuracy (%)	FLOPs (G)	Params (M)
FBANet		✓	✓	89.35	17.36	100.72
FBANet	✓	✓		87.55	0.72	15.71
FBANet	✓		✓	93.81	17.37	100.91
FBANet	✓	✓	✓	96.72	17.52	101.50

## Data Availability

Not applicable.

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
