# Peer review of "FBANet: Transfer Learning for Depression Recognition Using a Feature-Enhanced Bi-Level Attention Network"

_entropy, 2023, doi:10.3390/e25091350_

Round 1
Reviewer 1 Report
The authors present a novel approach to depression recognition based on sketches using deep learning methods. The authors manage to improve recognition accuracy up to 99.38%. The article may be of interest to specialists in the field of computer vision and medicine. However, there are a number of remarks about the work:
1) Line 45: duplication of physiological and physiological
2) Line 65: It is not good to start sentences by quoting [17-19].
3) I would strongly recommend that authors try approaches to image augmentation due to the small size of the dataset, and also mention such methods in the review [10.1109/SYNCHROINFO49631.2020.9166000, https://www.mdpi.com/2078-2489/11/ 2/125] and examine the results.
Reviewer 2 Report
Abstract should contain a statement about the challenges and limitations of current methods
In the Abstract, authors claim collection of primary data which is not very clear later in the Dataset Section.
If you collect new data, authors need to add detailed description of data collection protocols and Ethical information/process from the Institution or related body.
Writing style and readability can be improved throughout the manuscript
Number of contributions can be shrinked to 2 or maximum 3 as some are overlapping
Good to see highlighting the limitations of existing methods on page 2, I would recommend to include following recent literature related to Transfer learning and AI methods for facial expression, emotions etc., in Page 2 and Page 7 (Transfer learning) Sections: https://www.sciencedirect.com/science/article/pii/S0957417420310289
https://ieeexplore.ieee.org/abstract/document/10091555
Table and Figure Captions should include details
Line 283: How did you choose P and σ values?
Lot of Maths equations are well established and may not be added in the Method. Only can be referred to original work
How did author deal with the subjectivity while splitting the dataset in training/test sets? E.g., there can be multiple instances/sketches from same Subject/person where presence within the both training and testing partitions would lead to biased evaluation
Fig. 6 should be removed. Its not a technical report. You have reported outcomes in Tables.
Why your Recall is Higher than Precision? See Table 4. This shows biased performance
Highlight the limitations of your work at the end of Discussion Section.
see the comments above
Round 2
Reviewer 2 Report
While authors have responded most of the concerns, authors need to be very careful in responding Point 9.
- Subjectivity needs to be addressed. Authors can use e.g., Leave N out sampling to investigate the subjectivity . In other words, why authors do not mention how many participants are in total? Lets say X healthy participants draws 500 images with repetitions (as mentioned in original work), in case of your sampling method, 5 images from 1 participant can go to training while 2 images of same drawing from same participant can go to test set that will cause high Accuracy.
So just add a Table of results where images from same Participants (lets say 10 participants) are kept for Test while rest for train and see the outcomes.
- Secondly, I want to clearly see whether the Data augmentation is performed on Train and Test segments separately or entire data. In later choice, experiments are not appropriate and will produce high performance
na
Round 3
Reviewer 2 Report
NA
NA